# ON REPRESENTATION COMPLEXITY OF MODEL-BASED AND MODEL-FREE REINFORCEMENT LEARNING

**Hanlin Zhu**[*], **Baihe Huang**[*], **Stuart Russell**
EECS, UC Berkeley
{hanlinzhu,baihe_huang,russell}@berkeley.edu

## ABSTRACT

We study the representation complexity of model-based and model-free reinforcement learning (RL) in the context of circuit complexity. We prove theoretically that there exists a broad class of MDPs such that their underlying transition and reward functions can be represented by constant depth circuits with polynomial size, while the optimal $Q$-function suffers an exponential circuit complexity in constant-depth circuits. By drawing attention to the approximation errors and building connections to complexity theory, our theory provides unique insights into why model-based algorithms usually enjoy better sample complexity than model-free algorithms from a novel representation complexity perspective: in some cases, the ground-truth rule (model) of the environment is simple to represent, while other quantities, such as $Q$-function, appear complex. We empirically corroborate our theory by comparing the approximation error of the transition kernel, reward function, and optimal $Q$-function in various Mujoco environments, which demonstrates that the approximation errors of the transition kernel and reward function are consistently lower than those of the optimal $Q$-function. To the best of our knowledge, this work is the first to study the circuit complexity of RL, which also provides a rigorous framework for future research.

## 1 INTRODUCTIONS

In recent years, reinforcement learning (RL) has seen significant advancements in various real-world applications (Mnih et al., 2013; Silver et al., 2016; Moravčík et al., 2017; Shalev-Shwartz et al., 2016; Yurtsever et al., 2020; Yu et al., 2020; Kober et al., 2014). Roughly speaking, the RL algorithms can be categorized into two types: model-based algorithms (Draeger et al., 1995; Rawlings, 2000; Luo et al., 2018; Chua et al., 2018; Nagabandi et al., 2020; Moerland et al., 2023) and model-free algorithms (Mnih et al., 2015; Lillicrap et al., 2015; Van Hasselt et al., 2016; Haarnoja et al., 2018). Model-based algorithms typically learn the underlying dynamics of the model (i.e., the transition kernel and the reward function) and then learn the optimal policy utilizing the knowledge of the ground-truth model. On the other hand, model-free algorithms usually derive optimal policies through different quantities, such as $Q$-function and value function, without directly assimilating the underlying ground-truth models.

In statistical machine learning, the efficiency of an algorithm is usually measured by sample complexity. For the comparison of model-based and model-free RL algorithms, many previous works also focus on their sample efficiency gap, and model-based algorithm usually enjoys a better sample complexity than model-free algorihms (Jin et al., 2018; Zanette & Brunskill, 2019; Tu & Recht, 2019; Sun et al., 2019). In general, the error of learning [1] can be decomposed into three parts: optimization error, statistical error, and approximation error. Many previous efforts focus on optimization errors (Singh et al., 2000; Agarwal et al., 2020; Zhan et al., 2023) and statistical errors (Kakade, 2003; Strehl et al., 2006; Auer et al., 2008; Azar et al., 2017; Rashidinejad et al., 2022; Zhu & Zhang, 2024), while approximation error has been less explored. Although a line of previous work studies

---

[*]Equal contributions.
[1]For example, the performance difference between the learned policy and the optimal policy, which can be translated to sample complexity.

the sample complexity additionally caused by the approximation error under the assumption of a bounded model misspecification error (Jin et al., 2019; Wang et al., 2020; Zhu et al., 2023; Huang et al., 2021; Zhu et al., 2024), they did not take a further or deeper step to study for what types of function classes (including transition kernel, reward function, $Q$-function, etc.), it is reasonable to assume a small model misspecification error.

In this paper, we study approximation errors through the lens of representation complexities. Intuitively, if a function (transition kernel, reward function, $Q$-function, etc.) has a low representation complexity, it would be relatively easy to learn it within a function class of low complexity and misspecification error, thus implying a better sample complexity. The previous work Dong et al. (2020) studies a special class of MDPs with state space $\mathcal{S} = [0, 1]$, action space $\mathcal{A} = \{0, 1\}$, transition kernel piecewise linear with constant pieces and a simple reward function. By showing that the optimal $Q$-function requires an exponential number of linear pieces to approximate, they provide a concrete example that the $Q$-function has a much larger representation complexity than the transition kernel and reward function, which implies that model-based algorithms enjoy better sample complexity. However, the MDP class they study is restrictive. Thus, it is unclear whether it is a universal phenomenon that the underlying models have a lower representation complexity than other quantities such as $Q$-functions. Moreover, their metrics of measuring representation complexity, i.e., the number of pieces of piece-wise linear functions, is not fundamental, rigorous, or applicable to more general functions.

Therefore, we study representation complexity via circuit complexity (Shannon, 1949), which is a fundamental and rigorous metric that can be applied to arbitrary functions stored in computers. Also, since the circuit complexity has been extensively explored by numerous previous works (Shannon, 1949; Karp, 1982; Furst et al., 1984; Razborov, 1989; Smolensky, 1987; Vollmer, 1999; Leighton, 2014) and is still actively evolving, it could offer us a deep understanding of the representation complexity of RL, and any advancement of circuit complexity might provide new insights into our work. Theoretically, we show that there exists a general class of MDPs called Majority MDP (see more details in Section 3), such that their transition kernels and reward functions have much lower circuit complexity than their optimal $Q$-functions. This provides a new perspective for the better sample efficiency of model-based algorithms in more general settings. We also empirically validate our results by comparing the approximation errors of the transition kernel, reward function, and optimal $Q$-function in various Mujoco environments (see Section 4 for more details).

We briefly summarize our main contributions as follows:

- We are the first to study the representation complexity of RL under a circuit complexity framework, which is more fundamental and rigorous than previous works.

- We study a more general class of MDPs than previous work, demonstrating that it is common in a broad scope that the underlying models are easier to represent than other quantities such as $Q$-functions.

- We empirically validate our results in real-world environments by comparing the approximation error of the ground-truth models and $Q$-functions in various MuJoCo environments.

## 1.1 RELATED WORK

**Model-based v.s. Model-free algorithms.**    Many previous results imply that there exists the sample efficiency gap between model-based and model-free algorithms in various settings, including tabular MDPs (Strehl et al., 2006; Azar et al., 2017; Jin et al., 2018; Zanette & Brunskill, 2019), linear quadratic regulators (Dean et al., 2018; Tu & Recht, 2018; Dean et al., 2020), contextual decision processes with function approximation (Sun et al., 2019), etc. The previous work Dong et al. (2020) compares model-based and model-free algorithms through the lens of the expressivity of neural networks. They claim that the expressivity is a different angle from sample efficiency. On the contrary, in our work, we posit the representation complexity as one of the important reasons causing the gap in sample complexity.

**Ground-truth dynamics.**    The main result of our paper is that, in some cases, the underlying ground-truth dynamics (including the transition kernels and reward models) are easier to represent and thus learn, which inspires us to utilize the knowledge of the ground-truth model to boost the

learning algorithms. This is consistent with the methods of many previous algorithms that exploit the dynamics to boost model-free quantities (Buckman et al., 2018; Feinberg et al., 2018; Luo et al., 2018; Janner et al., 2019), perform model-based planning (Oh et al., 2017; Weber et al., 2017; Chua et al., 2018; Wang & Ba, 2019; Piché et al., 2018; Nagabandi et al., 2018; Du & Narasimhan, 2019) or improve the learning procedure via various other approaches (Levine & Koltun, 2013; Heess et al., 2015; Rajeswaran et al., 2016; Silver et al., 2017; Clavera et al., 2018).

**Approximation error.** A line of previous work (Jin et al., 2019; Wang et al., 2020; Zhu et al., 2023) study the sample complexity of RL algorithms in the presence of model misspecification. This bridges the connection between the approximation error and sample efficiency. However, these works directly assume a small model misspecification error without further justifying whether it is reasonable. Our results imply that assuming a small error of transition kernel or reward function might be more reasonable than $Q$-functions. Many other works study the approximation error and expressivity of neural networks (Bao et al., 2014; Lu et al., 2017; Dong et al., 2020; Lu et al., 2021). Instead, we study approximation error through circuit complexity, which provides a novel perspective and rigorous framework for future research.

**Circuit complexity.** Circuit complexity is one of the most fundamental concepts in the theory of computer science (TCS) and has been long and extensively explored (Savage, 1972; Valiant, 1975; Trakhtenbrot, 1984; Furst et al., 1984; Hastad, 1986; Smolensky, 1987; Razborov, 1987; 1989; Boppana & Sipser, 1990; Arora & Barak, 2009). In this work, we first introduce circuit complexity to reinforcement learning to study the representation complexity of different functions including transition kernel, reward function and $Q$-functions, which bridges an important connection between TCS and RL.

## 1.2 NOTATIONS

Let $\mathbf{1}_b = (1, \ldots, 1) \in \mathbb{R}^b$ denote the all-one vector and let $\mathbf{0}_b = (0, \ldots, 0) \in \mathbb{R}^b$ denote the all-zero vector. For any set $\mathcal{X}$ and any function $f : \mathcal{X} \to \mathcal{X}$, $f^{(k)}(x)$ is the value of $f$ applied to $x$ after $k$ times, i.e., $f^{(1)}(x) = f(x)$ and $f^{(k)}(x) = \underbrace{f(f(\ldots f(x))\ldots))}_{k}$.

Let $\lceil x \rceil$ denote the smallest integer greater than or equal to $x$, and let $\lfloor x \rfloor$ denote the greatest integer less than or equal to $x$. We use $\{0, 1\}^n$ to denote the set of $n$-bits binary strings. Let $\delta_x$ denote the Dirac measure: $\delta_x(A) = \begin{cases} 1, & x \in A \\ 0, & x \notin A \end{cases}$ and we use $\mathbb{1}(\cdot)$ to denote the indicator function. Let $[n]$ denote the set $\{1, 2, \ldots, n\}$ and let $\mathbb{N} = \{0, 1, 2, \ldots\}$ denote the set of all natural numbers.

## 2 PRELIMINARIES

### 2.1 MARKOV DECISION PROCESS

An episodic Markov Decision Process (MDP) is defined by the tuple $\mathcal{M} = (\mathcal{S}, \mathcal{A}, H, \mathbb{T}, r)$ where $\mathcal{S}$ is the state space, $\mathcal{A}$ is the action set, $H$ is the number of time steps in each episode, $\mathbb{T}$ is the transition kernel from $\mathcal{S} \times \mathcal{A}$ to $\Delta(\mathcal{S})$ and $r = \{r_h\}_{h=1}^H$ is the reward function. When $\mathbb{T}(\cdot|s, a) = \delta_{s'}$, i.e., $\mathbb{T}$ is deterministic, we also write $\mathbb{T}(s, a) = s'$. In each episode, the agent starts at a fixed initial state $s_1$ and at each time step $h \in [H]$ it takes action $a_h$, receives reward $r_h(s_h, a_h)$ and transits to $s_{h+1} \sim \mathbb{T}(\cdot|s_h, a_h)$. Typically, we assume $r_h(s_h, a_h) \in [0, 1]$.

A policy $\pi$ is a length-$H$ sequence of functions $\pi = \{\pi_h : \mathcal{S} \mapsto \Delta(\mathcal{A})\}_{h=1}^H$. Given a policy $\pi$, we define the value function $V_h^\pi(s)$ as the expected cumulative reward under policy $\pi$ starting from $s_h = s$ (we abbreviate $V^* := V_0^*$):

$$V_h^\pi(s) := \mathbb{E}\left[\sum_{t=h}^H r_t(s_t, a_t) \,\bigg|\, s_h = s, \pi\right]$$

and we define the $Q$-function $Q_h^\pi(s, a)$ as the the expected cumulative reward taking action $a$ in state $s_h = s$ and then following $\pi$ (we abbreviate $Q^* := Q_0^*$):

$$Q_h^\pi(s, a) := \mathbb{E}\left[\sum_{t=h}^H r_t(s_t, a_t) \,\middle|\, s_h = s, a_h = a, \pi\right].$$

The Bellman operator $\mathcal{T}_h$ applied to $Q$-function $Q_{h+1}$ is defined as follow

$$\mathcal{T}_h(Q_{h+1})(s, a) := r_h(s, a) + \mathbb{E}_{s' \sim \mathbb{T}(\cdot|s,a)}[\max_{a'} Q_{h+1}(s', a')].$$

There exists an optimal policy $\pi^*$ that gives the optimal value function for all states, i.e. $V_h^{\pi^*}(s) = \sup_\pi V_h^\pi(s)$ for all $h \in [H]$ and $s \in \mathcal{S}$ (see, e.g., Agarwal et al. (2019)). For notation simplicity, we abbreviate $V^{\pi^*}$ as $V^*$ and correspondingly $Q^{\pi^*}$ as $Q^*$. Therefore $Q^*$ satisfies the following Bellman optimality equations for all $s \in \mathcal{S}$, $a \in \mathcal{A}$ and $h \in [H]$:

$$Q_h^*(s, a) = \mathcal{T}_h(Q_{h+1}^*)(s, a).$$

## 2.2 FUNCTION APPROXIMATION

In value-based (model-free) function approximation, the learner is given a function class $\mathcal{F} = \mathcal{F}_1 \times \cdots \times \mathcal{F}_H$, where $\mathcal{F}_h \subset \{f : \mathcal{S} \times \mathcal{A} \mapsto [0, H]\}$ is a set of candidate functions to approximate the optimal Q-function $Q^*$.

In model-based function approximation, the learner is given a function class $\mathcal{F} = \mathcal{F}_1 \times \cdots \times \mathcal{F}_H$, where $\mathcal{F}_h \subset \{f : \mathcal{S} \times \mathcal{A} \mapsto \Delta(\mathcal{S})\}$ is a set of candidate functions to approximate the underlying transition function $\mathbb{T}$. Additionally, the learner might also be given a function class $\mathcal{R} = \mathcal{R}_1 \times \cdots \times \mathcal{R}_H$ where $\mathbb{R}_h \subset \{f : \mathcal{S} \times \mathcal{A} \mapsto [0, 1]\}$ is a set of candidate functions to approximate the reward function $r$. Typically, the reward function would be much easier to learn than the transition function.

To learn a function with a large representation complexity, one usually needs a function class with a large complexity to ensure that the ground-truth function is (approximately) realized in the given class. A larger complexity (e.g., log size, log covering number, etc.) of the function class would incur a larger sample complexity. Our main result shows that it is common that an MDP has a transition kernel and reward function with low representation complexity while the optimal $Q$-function has a much larger representation complexity. This implies that model-based algorithms might enjoy better sample complexity than value-based (model-free) algorithms.

## 2.3 CIRCUIT COMPLEXITY

To provide a rigorous and fundamental framework for representation complexity, in this paper, we use circuit complexity to measure the representation complexity.

Circuit complexity has been extensively explored in depth. In this section, we introduce concepts related to our results. One can refer to Arora & Barak (2009); Vollmer (1999) for more details.

**Definition 2.1** (Boolean circuits, adapted from Definition 6.1, (Arora & Barak, 2009)). For every $m, n \in \mathbb{Z}_+$, a *Boolean circuit $C$ with $n$ inputs and $m$ outputs* is a directed acyclic graph with $n$ sources and $m$ sinks (both ordered). All non-source vertices are called gates and are labeled with one of $\wedge$ (AND), $\vee$ (OR) or $\neg$ (NOT). For each gate, its *fan-in* is the number of incoming edges, and its *fan-out* is the number of outcoming edges. The *size* of $C$ is the number of vertices in it. The *depth* of $C$ is the length of the longest directed path in it. A *circuit family* is a sequence $\{C_n\}_{n \in \mathbb{Z}_+}$ of Boolean circuits where $C_n$ has $n$ inputs.

If $C$ is a Boolean circuit, and $x = (x_1, \ldots, x_n) \in \{0, 1\}^n$ is its input, then the output of $C$ on $x$, denoted by $C(x)$, is defined in the following way: for every vertex $v$ of $C$, a value $\mathrm{val}(v)$ is assigned to $v$ such that $\mathrm{val}(v)$ is given recursively by applying $v$'s logical operation on the values of the vertices pointed to $v$; the output $C(x)$ is a $m$-bits binary string $y = (y_1, \ldots, y_m)$, where $y_i$ is the value of the $i$-th sink.

**Definition 2.2** (Circuit computation). A *circuit family* is a sequence $C = (C_1, C_2, \ldots, C_n, \ldots)$, where for every $n \in \mathbb{Z}_+$, $C_n$ is a Boolean circuit with $n$ inputs. Let $f_n$ be the function computed by $C_n$. Then we say that $C$ *computes the function* $f : \{0, 1\}^* \to \{0, 1\}^*$, which is defined by

$$f(w) := C_{|w|}(w), \ \forall w \in \{0, 1\}^*$$

where $|w|$ is the bit length of $w$. More generally, we say that a function $f : \mathbb{N} \to \mathbb{N}$ can be computed by $C$ if $f$ can be computed by $C$ where the inputs and the outputs are represented by binary numbers.

**Definition 2.3** ($(k, m)$-DNF). A $(k, m)$-DNF is a disjunction of conjuncts, i.e., a formula of the form

$$\bigvee_{i=1}^{n} \left( \bigwedge_{j=1}^{k_i} X_{i,j} \right)$$

where $k_i \leq k$ for every $i \in [n]$, $\sum_{i=1}^{n} k_i \leq m$, and $X_{i,j}$ is either a primitive Boolean variable or the negation of a primitive Boolean variable.

**Definition 2.4** ($\mathbf{AC}^0$). $\mathbf{AC}^0$ is the class of all boolean functions $f : \{0, 1\}^* \to \{0, 1\}^*$ for which there is a circuit family with unbounded fan-in, $n^{O(1)}$ size, and constant depth that computes $f$.

In this paper, we study representation complexity within the class of constant-depth circuits. Our results depend on two "hard" functions, parity and majority functions, which require exponential size circuits to compute and thus are not in $\mathbf{AC}^0$. Below, we formally define these two functions respectively.

**Definition 2.5** (Parity). For every $n \in \mathbb{Z}_+$, the *n-variable parity function* $\mathsf{PARITY}_n : \{0, 1\}^n \to \{0, 1\}$ is defined by $\mathsf{PARITY}_n(x_1, \ldots, x_n) = \sum_{i=1}^{n} x_i \mod 2$. The *parity function* $\mathsf{PARITY} : \{0, 1\}^* \to \{0, 1\}$ is defined by

$$\mathsf{PARITY}(w) := \mathsf{PARITY}_{|w|}(w), \; \forall w \in \{0, 1\}^*.$$

**Proposition 2.6** ((Furst et al., 1984)). $\mathsf{PARITY} \notin \mathbf{AC}^0$.

## 3 THEORETICAL RESULTS

We show our main results in this section that there exists a broad class of MDPs that demonstrates a separation between the representation complexity of the ground-truth model and that of the value function. This reveals one of the important reasons that model-based algorithms usually enjoy better sample efficiency than value-based (model-free) algorithms.

The previous work Dong et al. (2020) also conveyed similar messages. However, they only study a very special MDP while we study a more general class of MDPs. Moreover, Dong et al. (2020) measures the representation complexity of functions by the number of pieces for a piecewise linear function, which is not rigorous and not applicable to more general functions.

To study the representation complexity of MDPs under a more rigorous and fundamental framework, we introduce circuit complexity to facilitate the study of representation complexity. In this work, we focus on circuits with constant-depth. We first show a warm-up example in Section 3.1, and then extend our results to a broader class of MDPs in Section 3.2. In Appendix E, we also briefly discuss possible extensions to POMDP and stochastic MDP.

### 3.1 WARM UP EXAMPLE

In this section we show a simple MDP that demonstrates a separation between the circuit complexity of the model function and that of the value function.

**Definition 3.1** (Parity MDP). An *n-bits parity MDP* is defined as follows: the state space is $\{0, 1\}^n$, the action space is $\{(i, j) : i, j \in [n]\}$, and the planning horizon is $H = n$. Let the reward function be defined by: $r(s, a) = \begin{cases} 1, & s = \mathbf{0}_n \\ 0, & \text{otherwise} \end{cases}$. Let the transition function be defined as follows: for each state $s$ and action $a = (i, j)$, transit with probability 1 to $s'$ where $s'$ is given by flipping the $i$-th and $j$-th bits of $s$.

Both $\mathbb{T}$ and $r$ can be computed by a circuit with polynomial-size and constant depth. Indeed, consider the following circuit:

$$C_{\text{reward}}(s) = s_1 \wedge s_2 \wedge \cdots \wedge s_n.$$

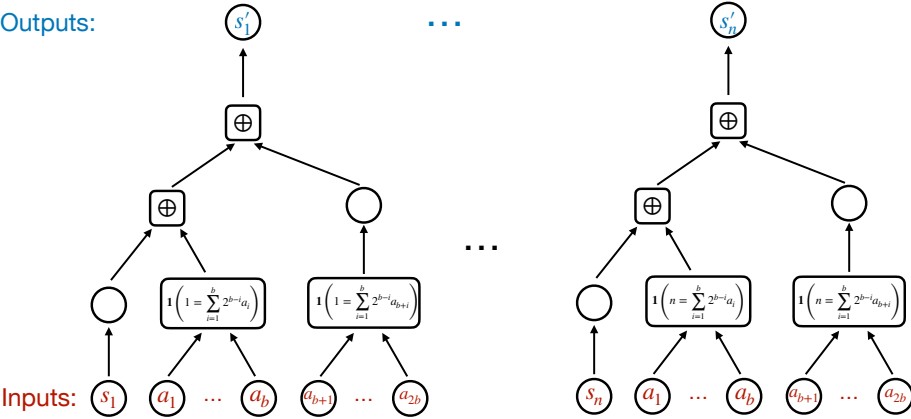

Figure 1: Constant-depth circuit of the model transition function in parity MDP. An empty node is directly assigned the value of the node pointing to it.

It can be verified that $C_{\text{reward}} = r$ and it has size $n$. For the model transition function, we consider the binary representation of the action: for each $a = (i, j)$, let $(a_1, \ldots, a_b)$ denote the binary representation of $i$ and let $(a_{b+1}, \ldots, a_{2b})$ denote the binary representation of $j$, where $b = \lceil \log(n + 1) \rceil$. Then define the following circuit:

$$C_{\text{model}}(s, a) = (s_k \oplus \delta_k(a_1, \ldots, a_b) \oplus \delta_k(a_{b+1}, \ldots, a_{2b}))_{k=1}^n$$

where $\oplus$ is the XOR gate and $\delta_k(x_1, \ldots, x_b) = \mathbb{1}\left(k = \sum_{i=1}^b 2^{b-i} \cdot x_i\right)$. We visualize this circuit in Figure 1. Since the XOR gate and the gate $\delta_k$ can all be implemented by binary circuits with polynomial size and constant depth (notice that $a \oplus b = (\neg a \vee b) \wedge (a \vee \neg b)$ and $\delta_k(x_1, \ldots, x_b) = (x_1 \vee k_1) \vee \cdots \vee (x_b \vee k_b)$ where $k_1 k_2 \cdots k_b$ is the binary representation of $k$), $C_{\text{model}}$ also has polynomial size and constant depth.

However, the optimal $Q$-function $Q^*$ can not be computed by a circuit with polynomial size and constant depth. Indeed, it suffices to see that the value function $V^*$ can not be computed by a circuit with polynomial size and constant depth. To see this, notice that $\mathbb{1}(V^* > 0)$ is the parity function, since if there are even numbers of 1's in $(s_1, \ldots, s_n)$, then there always exists a sequence of actions to transit to the reward state; otherwise, the number of 1's remains odd and will never become $\mathbf{0}_n$. Suppose for the sake of contradiction that there exists a circuit $C_{\text{value}}$ with polynomial size and constant depth that computes $V^*$, then the circuit

$$C_{\text{PARITY}}(s, a) = C_{\text{value}}(s, a)_1 \vee C_{\text{value}}(s, a)_2 \vee \cdots \vee C_{\text{value}}(s, a)_n$$

computes the parity function for $s$ and belongs to the class $\mathbf{AC}^0$. This contradicts Proposition 2.6.

## 3.2 A BROADER FAMILY OF MDPs

In this section, we present our main results, i.e., there exists a general class of MDP, of which nearly all instances of MDP have low representation complexity for the transition model and reward function but suffer an exponential representation complexity for the optimal $Q$-function.

We consider a general class of MDPs called 'majority MDP', where the states are comprised of representation bits that reflect the situation in the underlying environment and control bits that determine the transition of the representation bits. We first give the definition of majority MDP and then provide intuitive explanations.

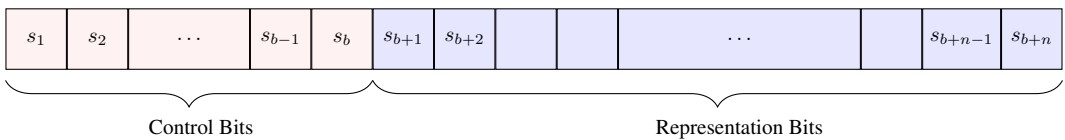

Figure 2: Illustration of state for majority MDPs.

**Definition 3.2** (Control function). We say that a map $f$ from $\{0,1\}^b$ to itself is a *control function over* $\{0,1\}^b$, if $f(\mathbf{1}_b) = \mathbf{1}_b$, and

$$\left\{ f^{(k)}(\mathbf{0}_b) : k = 1, 2, \ldots, 2^b - 1 \right\} = \{0,1\}^b \backslash \{\mathbf{0}_b\}.$$

**Definition 3.3** (Majority MDP). An *n-bits majority MDP* with reward state $s_{\text{reward}} \in \{0,1\}^n$, control function $f : \{0,1\}^{\lceil \log(n+1) \rceil} \to \{0,1\}^{\lceil \log(n+1) \rceil}$, and condition $C : \{0,1\}^n \to \{0,1\}$ is defined by the following:

- The state space is given by $\{0,1\}^{n+b}$ where $b = \lceil \log(n+1) \rceil$. For convenience, we assume $n = 2^b - 1$. Each state is comprised of two subparts $s = (s[c], s[r])$, where $s[c] = (s_1, \ldots, s_b) \in \{0,1\}^b$ is called the control bits, and $s[r] = (s_{b+1}, \ldots, s_{b+n}) \in \{0,1\}^n$ is called the representation bits (see Figure 2);

- The action space is $\{0,1\}$; The planning horizon is $H = 2^b + n$;

- The reward function is defined by: $r(s, a) = \begin{cases} 1, & s[r] = s_{\text{reward}}, s[c] = \mathbf{1}_b \\ 0, & \text{otherwise} \end{cases}$;

- The transition function $\mathbb{T}$ is defined as follows: define the flipping function $g : \{0,1\}^n \times [n] \to \{0,1\}^n$ by $g(s, i) = (s_1, \ldots, \neg s_i, \ldots, s_n)$, then $\mathbb{T}$ is given by:

$$\mathbb{T}(s, a) = \begin{cases} \left( s[c], g(s[r], \sum_{j=1}^{b} 2^{b-j} \cdot s_j) \right), & a = 1, C(s[r]) = 1 \\ (f(s[c]), s[r]), & \text{otherwise} \end{cases}$$

  that is, if $a = 1$ and $C(s[r]) = 1$, then transit to $s' = (s[c], g(s[r], i))$ where $i = \sum_{j=1}^{b} 2^{b-j} \cdot s_j$[2] and $g_i(s[r])$ is given by flipping the $i$-th bits of $s[r]$ (keep the control bits while flip the $i$-th coordinate of the representation bits); otherwise transit to $s' = (f(s[c]), s[r])$ (keep the representation bits and apply the control function $f$ to the control bits).

When $C \equiv 1$, we call such an MDP unconditioned.

Although many other MDPs lie outside of the Majority MDP class, most are too random to become meaningful (for example, the MDP where the reward state is in a $o(n)$-sized connected component). Thus, instead of studying a more general class of MDPs, we consider one representative class of MDPs and separate three fundamental notions in RL: control, representation, and condition. We elaborate on these aspects in the following remark.

**Remark 3.4** (Control function and control bits). *In Majority MDPs, the control bits start at $\mathbf{0}_b$ and traverse all b-bits binary strings before ending at $\mathbf{1}_b$. This means that the agent can can flip every entry of the representation bits, and therefore, the agent is able to change the representation bits to any n-bits binary string within $2^b + n$ time steps in unconditional settings.*

*The control function is able to express any ordering of b-bits strings (starting from $\mathbf{0}_b$ and ending with $\mathbf{1}_b$) in which the control bits are taken. With this expressive power, the framework of Majority MDP simplifies the action space to fundamental case of $\mathcal{A} = \{0,1\}$.*

**Remark 3.5** (Representation bits). *In general, the states of any MDP (even for MDP with continuous state space) are stored in computer systems in finitely many bits. Therefore, we allocate n bits in the representation bits to encapsulate and delineate the various states within an MDP.*

---

[2]That is, $s[c]$ as a binary number equals $i$.

**Remark 3.6** (Condition function). *The condition function simulates and expresses the rules of transition in an MDP. In many real-world applications, the decisions made by an agent only take effect (and therefore cause state transition) under certain underlying conditions, or only enable transitions to certain states that satisfy the conditions: for example, an marketing maneuver made by a company will only make an influence if it observes the advertisement law and regulations; a move of a piece chess must follow the chess rules; a treatment decision can only affect some of the measurements taken on an individual; a resource allocation is subject to budget constraints; etc.*

Finally, the following two theorems show the separation result of circuit complexity between the model and the optimal $Q$-function for Majority MDP in both unconditional and conditional settings.

**Theorem 3.7** (Separation result of majority MDP, unconditioned setting). *For any reward state $s_{\mathrm{reward}} \in \{0,1\}^n$ and any control function $f : \{0,1\}^b \to \{0,1\}^b$, the unconditioned $n$-bits majority MDP with reward state $s_{\mathrm{reward}}$, control function $f$ has the following properties:*

- *The reward function and the transition function can be computed by circuits with polynomial (in $n$) size and constant depth.*

- *The optimal $Q$-function (at time step $t = 0$) cannot be computed by a circuit with polynomial size and constant depth.*

**Theorem 3.8** (Separation result of majority MDP, conditioned setting). *Fix $m < n/2$. Let $\rho$ be uniform distribution over $(O(1), m)$-DNFs of $n$ Boolean variables. Then for any reward state $s_{\mathrm{reward}} \in \{0,1\}^n$ and any control function $f : \{0,1\}^b \to \{0,1\}^b$, with probability at least $1 - e^{-\Omega(m)}$, the $n$-bits majority MDP with reward state $s_{\mathrm{reward}}$, control function $f$, and condition $C$ sampled from $\rho$, has the following properties:*

- *The reward function and the transition function can be computed by circuits with polynomial (in $n$) size and constant depth.*

- *The optimal $Q$-function (at time step $t = 0$) cannot be computed by a circuit with polynomial size and constant depth.*

The proof of Theorem 3.7 and Theorem 3.8 are deferred to Appendix B.3 and Appendix B.2. In short, they imply that $\mathbb{T}, r \in \mathbf{AC}_0$ and $Q^* \notin \mathbf{AC}_0$. In fact, we show that the value functions cannot be computed by a circuit with polynomial size and constant depth. Due to the relationship $Q^*(s, a) = r(s, a) + V^*(\mathbb{T}(s, a))$, we will treat these two functions synonymously and refer to them collectively as the "value function."

## 4 EXPERIMENTS

Theorems 3.7 and 3.8 indicate that there exists a broad class of MDPs in which the transition functions and the reward functions have much lower circuit complexity than the optimal $Q$-functions (actually also value functions according to our proof for Majority MDPs). This observation, therefore, implies that value functions might be harder to approximate than transition functions and the reward functions, and gives rise to the following question:

*In general MDPs, are the value functions harder to approximate than transition functions and the reward functions?*

In this section, we seek to answer this question via experiments on common simulated environments. Specifically, we fix $d, m \in \mathbb{Z}_+$, and let $\mathcal{F}$ denote the class of $d$-depth (i.e., $d$ hidden layers), $m$-width neural networks (with input and output dimensions tailored to the context). The quantities of interest are the following relative approximation errors

$$e_{\mathrm{model}} = \min_{P \in \mathcal{F}} \frac{\mathbb{E}[\|P(s, a) - \mathbb{T}(s, a)\|^2]}{\mathbb{E}[\|\mathbb{T}(s, a)\|^2]}$$

$$e_{\mathrm{reward}} = \min_{R \in \mathcal{F}} \frac{\mathbb{E}[(R(s, a) - r(s, a))^2]}{\mathbb{E}[(r(s, a))^2]}$$

$$e_{Q\text{-function}} = \min_{Q \in \mathcal{F}} \frac{\mathbb{E}[(Q(s, a) - Q^*(s, a))^2]}{\mathbb{E}[(Q^*(s, a))^2]}$$

where the expectation is over the distribution of the optimal policy and the mean squared errors are divided by the second moment so that the scales of different errors will match. Therefore, $e_{\text{model}}, e_{\text{reward}}, e_{\text{Q-function}}$ stand for the difficulty for a $d$-depth, $m$-width neural networks to approximate the transition function, the reward function, and the $Q$-function, respectively.

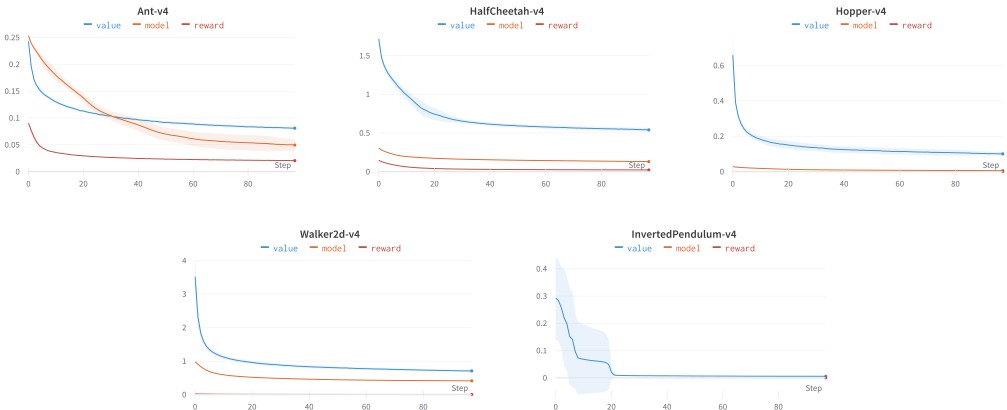

Figure 3: Approximation errors of the optimal $Q$-functions, reward functions, and transition functions in MuJoCo environments. In each environment, we run 5 independent experiments and report the mean and standard deviation of the approximation errors. All curves (as well as those in Figure 4-9) are displayed with an exponential average smoothing with rate 0.2.

For common MuJoCo Gym environments (Brockman et al., 2016), including Ant-v4, Hopper-v4, HalfCheetah-v4, InvertedPendulum-v4, and Walker2d-v4 , we find these objectives by training $d$-depth, $m$-width neural networks to fit the corresponding values over the trajectories generated by an SAC-trained agent. [3] In Figure 3, we visualize[4] the approximation errors under $d = 2, w = 32$. Among the approximation objectives, the reward functions and the model transition functions are accessible, and we use Soft-Actor-Critic (Haarnoja et al., 2018) to learn the optimal $Q$-function. We consistently observe that the approximation errors of the optimal $Q$-function, in all environments, are much greater than the approximation errors of the transition and reward function. This finding concludes that in the above environments, the optimal $Q$-functions are more difficult to approximate than the transition functions and the reward function, which partly consolidate our hypothesis. We provide experimental details in Appendix C and additional experiments in Appendix D.

## 5 CONCLUSIONS

In this paper, we find that in a broad class of Markov Decision Processes, the transition function and the reward function can be computed by constant-depth, polynomial-sized circuits, whereas the optimal $Q$-function requires an exponential size for constant-depth circuits to compute. This separation reveals a rigorous gap in the representation complexity of the $Q$-function, the reward, and the transition function. Our experiments further corroborate that this gap is prevalent in common real-world environments.

Our theory lays the foundation of studying the representation complexity in RL and raises several open questions:

1. If we randomly sample an MDP, does the separation that the value function $\notin \mathbf{AC}^0$ and the reward and transition function $\in \mathbf{AC}^0$ occurs in high probability?

2. For $k \geq 1$, are there broad classes of MDPs such that the value function $\notin \mathbf{AC}^k$ and the reward and transition function $\in \mathbf{AC}^k$?

3. What are the typical circuit complexities of the value function, the reward function, and transition function?

---

[3]Our code is available at `https://github.com/realgourmet/rep_complexity_rl`.

[4]We used Weights & Biases (Biewald, 2020) for experiment tracking and visualizations.

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

## A  Supplementary Background on Circuit Complexity

**Definition A.1** (Majority). For every $n \in \mathbb{Z}_+$, the *n-variable majority function* $\mathsf{MAJORITY}_n : \{0,1\}^n \to \{0,1\}$ is defined by $\mathsf{MAJORITY}_n(x_1, \ldots, x_n) = \mathbb{1}(\sum_{i=1}^n x_i > n/2)$. The *majority function* $\mathsf{MAJORITY} : \{0,1\}^* \to \{0,1\}$ is defined by

$$\mathsf{MAJORITY}(w) := \mathsf{MAJORITY}_{|w|}(w), \ \forall w \in \{0,1\}^*.$$

**Proposition A.2** ((Razborov, 1987; Smolensky, 1987; 1993)). $\mathsf{MAJORITY} \notin \mathbf{AC}^0$.

We also introduce another function that can be represented by polynomial-size constant-depth circuits, contrary to the above two "hard" functions.

**Definition A.3** (Addition). For every $n \in \mathbb{Z}_+$, the *length-n integer addition function* $\mathsf{ADDITION}_n : \{0,1\}^n \times \{0,1\}^n \to \{0,1\}^{n+1}$ is defined as follows: for any two binary strings $a_1, a_2 \in \{0,1\}^n$, the value $\mathsf{ADDITION}_n(a_1, a_2)$ is the $(n+1)$-bits binary representation of $a_1 + a_2$. The integer addition function $\mathsf{ADDITION} : \{0,1\}^* \to \{0,1\}^*$ is defined by

$$\mathsf{ADDITION}(w_1, w_2) := \mathsf{ADDITION}_{\max\{|w_1|, |w_2|\}}(w_1, w_2), \ \forall w_1, w_2 \in \{0,1\}^*.$$

**Proposition A.4** (Proposition 1.15, (Vollmer, 1999)). $\mathsf{ADDITION} \in \mathbf{AC}^0$.

**Definition A.5** (Maximum). For every $n \in \mathbb{Z}_+$, the *length-n maximum function* $\mathsf{MAX}_n : \{0,1\}^n \times \{0,1\}^n \to \{0,1\}^n$ is defined as follows: for any two binary strings $a_1, a_2 \in \{0,1\}^n$, the value $\mathsf{MAX}_n(a_1, a_2)$ is the *n*-bits binary representation of $\max\{a_1, a_2\}$. The maximum function $\mathsf{MAX} : \{0,1\}^* \to \{0,1\}^*$ is defined by

$$\mathsf{MAX}(w_1, w_2) := \mathsf{MAX}_{\max\{|w_1|, |w_2|\}}(w_1, w_2), \ \forall w_1, w_2 \in \{0,1\}^*.$$

**Proposition A.6.** $\mathsf{MAX} \in \mathbf{AC}^0$.

*Proof of Proposition A.6.* Fix any $n \in \mathbb{Z}_+$, and assume $w_1, w_2 \in \{0,1\}^n$. Let

$$k = \bigvee_{i=1}^n \left( (w_1)_i \wedge \neg(w_2)_i \wedge \left( \bigwedge_{j=1}^{i-1} \neg((w_1)_j \oplus (w_2)_j) \right) \right).$$

Therefore, $k = \mathbb{1}(w_1 > w_2)$. Let the $i$-th bit of output be

$$((w_1)_i \wedge k) \vee ((w_2)_i \wedge \neg k),$$

which is exact the $i$-th bit of $\max\{w_1, w_2\}$. This circuit has polynomial size and constant depth, which completes the proof. $\square$

## B  Proofs of Main Results

### B.1  Useful results

We will use the following results frequently in the proofs.

**Claim B.1.** If $C_0, C_1, \ldots, C_n$ are circuits with polynomial (in $n$) size and constant depth, where $C_i$ has $k_i$ outputs for each $i \in [n]$ and $C_0$ has $\sum_{i=1}^n k_i$ inputs, then the circuit

$$C_0\left(C_1(x_1), \ldots, C_n(x_n)\right)$$

also has polynomial (in $n$) size and constant depth.

**Claim B.2.** The XOR gate can be computed by a circuit with constant size.

**Claim B.3.** For any integer $k = \mathrm{poly}(n)$, the gate $\delta(x_1, \ldots, x_n) = \mathbb{1}\left(k = \sum_{i=1}^n 2^{n-i} \cdot x_i\right)$ can be computed by a circuit with polynomial size and constant depth.

### B.2 PROOF OF THEOREM 3.7

We need the following lemma.

**Lemma B.4** (Property of value function in unconditioned Majority MDPs). *In an unconditioned $n$-bits majority MDP with reward state $s_{\text{reward}}$ and control function $f$, the value function $V^*$ (at time step $t = 0$) over $\{s \in \{0,1\}^{b+n} : s[c] = \mathbf{0}_b\}$ is given by the following:*

$$V^*(s) = n - \sum_{i=1}^{n}(\neg(s_{b+i} \oplus s_{\text{reward},i})) + 1.$$

*Proof.* To find $V^*(s)$, it suffices to count the number of time steps an optimal agent takes to reach the reward state $(\mathbf{1}_b, s_{\text{reward}})$.

For the control bits to traverse from $\mathbf{0}_b$ to $\mathbf{1}_b$, it takes $2^b - 1$ time steps. Indeed, by Definition 3.2, $f^{(i)}(\mathbf{0}_b) \neq \mathbf{1}_b$ for any $i < 2^b - 1$ and $f^{(2^b-1)}(\mathbf{0}_b) = \mathbf{1}_b$ (otherwise, $\left|\{f^{(k)}(\mathbf{0}_b) : k = 1, 2, \ldots, 2^b - 1\}\right| < 2^b - 1$ and as a result $\{f^{(k)}(\mathbf{0}_b) : k = 1, 2, \ldots, 2^b - 1\}$ can not traverse $\{0,1\}^b \backslash \{\mathbf{0}_b\}$). Thus, this corresponds to at least $2^b - 1$ time steps in which action $a = 0$ is played.

For the representation bits, starting from $s[r]$, reaching $s_{\text{reward}}$ takes at least

$$\sum_{i=1}^{n}(\neg(s_{b+i} \oplus s_{\text{reward},i}))$$

number of flipping, because each index $i$ such that $s_{b+i} \neq s_{\text{reward},i}$ needs to be flipped. This corresponds to at least $\sum_{i=1}^{n}(\neg(s_{b+i} \oplus s_{\text{reward},i}))$ time steps in which action $a = 1$ is played.

In total, the agent needs to take at least $2^b + \sum_{i=1}^{n}(\neg(s_{b+i} \oplus s_{\text{reward},i})) - 1$ times steps before it can receive positive rewards. Since there are $2^b + n$ time steps in total, it follows that the agent gets a positive reward in at most $n - \sum_{i=1}^{n}(\neg(s_{b+i} \oplus s_{\text{reward},i})) + 1$ time steps. As a consequence,

$$V^*(s) \leq n - \sum_{i=1}^{n}(\neg(s_{b+i} \oplus s_{\text{reward},i})) + 1$$

On the other hand, consider the following policy:

$$\pi^*(s) = \begin{cases} 1, & \text{if } s_{b+i} \neq s_{\text{reward},i}, \text{ where } i = \sum_{j=1}^{b} 2^{b-j} \cdot s_j \\ 0, & \text{otherwise.} \end{cases}$$

This policy reaches the reward state in $2^b + \sum_{i=1}^{n}(\neg(s_{b+i} \oplus s_{\text{reward},i})) - 1$ time steps. Indeed, the control bits takes $2^b - 1$ times steps to reach $\mathbf{1}_b$. During these time steps, the control bits (as a binary number) traverses $0, \ldots, 2^b - 1$. Thus for any $i \in [n]$ such that $s_{b+i} \neq s_{\text{reward},i}$, there exists a time $t$ such that the control bits at this time $t$ (as a binary number) equals $i$, and at this time step, the agent flipped the $i$-th coordinate of the representation bits by playing action 1. Therefore, the representation bits takes $\sum_{i=1}^{n}(\neg(s_{b+i} \oplus s_{\text{reward},i}))$ times steps to reach $s_{\text{reward}}$. Combining, the agent arrives the state $(\mathbf{0}_b, s_{\text{reward}})$ after $2^b + \sum_{i=1}^{n}(\neg(s_{b+i} \oplus s_{\text{reward},i})) - 1$ time steps, and then collects reward 1 in each of the remaining $n - \sum_{i=1}^{n}(\neg(s_{b+i} \oplus s_{\text{reward},i})) + 1$ time steps, resulting in a value of

$$V^*(s) = n - \sum_{i=1}^{n}(\neg(s_{b+i} \oplus s_{\text{reward},i})) + 1.$$

$\square$

**Lemma B.5.** *Any control function can be computed by a constant depth, polynomial sized (in $n$) circuit.*

*Proof.* By Claim 2.13 in Arora & Barak (2009), any Boolean function can be computed by a CNF formula, i.e., Boolean circuits of the form

$$\bigwedge_{i=1}^{n}\left(\bigvee_{j=1}^{k_i} X_{i,j}\right).$$

As result, the control function can be computed by a depth-2, $2^b = \text{poly}(n)$ sized circuit. $\square$

Now we return to the proof of Theorem 3.7.

*Proof.* First, we show that the model transition function can be computed by circuits with polynomial size and constant depth. Consider the following circuit:

$$C_{\text{model}}(s, a) = (g(s_1, \ldots, s_b, a), (s_{b+k} \oplus (\delta_k(s_1, \ldots, s_b) \wedge a))_{k=1}^n)$$

where $\oplus$ is the XOR gate, $g : \{0,1\}^{b+1} \to \{0,1\}^b$ such that $g(x_1, \ldots, x_b, a)_i = (f(x_1, \ldots, x_b)_i \wedge \neg a) \vee (x_i \wedge a)$ for all $i \in [b]$, and $\delta_k(x_1, \ldots, x_b) = \mathbb{1}(k = \sum_{j=1}^b 2^{b-j} \cdot x_j)$. We can verify that $C_{\text{model}} = \mathbb{T}$. By Claim B.2, Claim B.3, and Lemma B.5, the XOR gate, the control function $f$, and the gate $\delta_k$ can all be implemented by binary circuits with polynomial size and constant depth. As a result of Claim B.1, $C_{\text{model}}$ also has polynomial size and constant depth.

Now, the reward function can be computed by the following simple circuit:

$$C_{\text{reward}}(s) = s_1 \wedge \ldots \wedge s_b \wedge (\neg(s_{b+1} \oplus s_{\text{reward},1})) \wedge \cdots \wedge (\neg(s_{b+n} \oplus s_{\text{reward},n})).$$

Finally, we show that the value function can not be computed by a circuit with constant depth and polynomial size. By Lemma B.4, we have

$$n + 1 - V^*(s) = \sum_{i=1}^n (\neg(s_{b+i} \oplus s_{\text{reward},i})) \in \{0\} \cup [n] = \{0\} \cup [2^b - 1],$$

which can be represented in binary form with $b$ bits. Therefore, the first (most significant) bit of $n + 1 - V^*(s)$ is the majority function of $(\neg(s_{b+i} \oplus s_{\text{reward},i}))_{i=1}^n$. If there exists a circuit $C_{\text{value}} = V^*$ with polynomial size and constant depth, then the circuit defined by

$$C_{\text{MAJORITY}}(x_1, \ldots, x_n) = n + 1 - C_{\text{value}}(\mathbf{0}_b, \neg x_1 \oplus s_{\text{reward},1}, \ldots, \neg x_n \oplus s_{\text{reward},n})$$

outputs the majority function in the first bit. By Proposition A.4, $C_{\text{MAJORITY}}$ also has polynomial size and constant depth, which contradicts Proposition 3.8. This means that $V^*$ can not be computed by circuits with polynomial size and constant depth. Notice that $V^*(s) = \max_{a \in \mathcal{A}} Q^*(s, a) = \max\{Q^*(s, 0), Q^*(s, 1)\}$. By Proposition A.6 and Claim B.1, we conclude that the optimal $Q$-function can not be computed by circuits with polynomial size and constant depth. $\square$

## B.3 PROOF OF THEOREM 3.8

We need the following lemmata.

**Lemma B.6.** *With probability at least* $1 - e^{-\Omega(m)}$, *there exists a set* $A \subset [n]$ *such that* $|A| \geq n/2$ *and* $C(x) = 1$ *holds for any binary string* $x \in \{0,1\}^n$ *satisfying* $x_i = s_{\text{reward},i}, \forall i \notin A$.

*Proof.* It suffices to show that $C(s_{\text{reward}}) = 1$ with at least $1 - e^{-\Omega(m)}$. Indeed, in this case, since $m < n/2$, $C$ is independent on at least $n/2$ of the variables $x_1, \ldots, x_n$. The indices of such variables form the set $A$ that we are looking for.

To show that $C(s_{\text{reward}}) = 1$ with at least $1 - e^{-\Omega(m)}$, we assume WLOG that

$$C(x_1, \ldots, x_n) = \bigvee_{i=1}^n \left( \bigwedge_{j=1}^{k_i} X_{i,j} \right)$$

where $X_{i,j} = x_l$ or $X_{i,j} = \neg x_l$ for some $l \in [m]$. Notice

$$\mathbb{P}\left( \left( \bigwedge_{j=1}^{k_i} X_{i,j} \right) = 1 \right) = 2^{-k_i} = \Omega(1)$$

since each $X_{i,j}$ is sampled i.i.d. from $\{x_l, \neg x_l\}_{l \in [m]}$ uniformly at random. It follows that

$$\mathbb{P}(C(x_1, \ldots, x_n) = 1) = 1 - \prod_{i=1}^n \mathbb{P}\left( \left( \bigwedge_{j=1}^{k_i} X_{i,j} \right) = 0 \right)$$

$$= 1 - \prod_{i=1}^n (1 - 2^{-k_i})$$

$$\geq 1 - e^{-\Omega(m)}.$$

$\square$

**Lemma B.7** (Property of value function in conditioned Majority MDPs). *In an $n$-bits majority MDP with reward state $s_{\text{reward}}$, control function $f$ and condition $C$, if there exists a set $A \subset [n]$ such that $|A| = \lceil n/2 \rceil = 2^{b-1} - 1$ and $C(x) = 1$ holds for any binary string $x \in \{0,1\}^n$ satisfying $x_i = s_{\text{reward},i}, \forall i \notin A$, then the value function $V^*$ over $\{s \in \{0,1\}^{b+n} : s[c] = \mathbf{0}_b, s_{b+i} = s_{\text{reward},i} \ (\forall i \notin A)\}$ is given by the following:*

$$V^*(s) = n - \sum_{i \in A}(\neg(s_{b+i} \oplus s_{\text{reward},i})) + 1.$$

*Proof.* To find $V^*(s)$, it suffices to count the number of actions it takes to reach the reward state $(\mathbf{1}_b, s_{\text{reward}})$.

For the control bits to travel from $\mathbf{0}_b$ to $\mathbf{1}_b$, it takes $2^b - 1$ time steps. Indeed, by Definition 3.2, $f^{(i)}(\mathbf{0}_b) \neq \mathbf{1}_b$ for any $i < 2^b - 1$ and $f^{(2^b-1)}(\mathbf{0}_b) \neq \mathbf{1}_b$ (otherwise, $\left|\{f^{(k)}(\mathbf{0}_b) : k = 1, 2, \ldots, 2^b - 1\}\right| < 2^b - 1$ and as a result $\{f^{(k)}(\mathbf{0}_b) : k = 1, 2, \ldots, 2^b - 1\}$ can not traverse $\{0,1\}^b \backslash \{\mathbf{0}_b\}$). Thus, this corresponds to at least $2^b - 1$ time steps in which action $a = 0$ is played.

For the representation bits, starting from $s[r]$, reaching $s_{\text{reward}}$ takes at least

$$\sum_{i \in A}(\neg(s_{b+i} \oplus s_{\text{reward},i}))$$

number of flipping, as each index $i$ such that $s_{b+i} \neq s_{\text{reward},i}$ needs to be flipped. This corresponds to at least $\sum_{i \in A}(\neg(s_{b+i} \oplus s_{\text{reward},i}))$ time steps in which action $a = 1$ is played.

In total, the agent needs to take at least $2^b + \sum_{i \in A}(\neg(s_{b+i} \oplus s_{\text{reward},i})) - 1$ times steps before it can receive positive rewards. Since there are $2^b + n$ time step in total, it follows that the agent gets positive reward in at most $n - \sum_{i \in A}(\neg(s_{b+i} \oplus s_{\text{reward},i})) + 1$ time steps. As a consequence,

$$V^*(s) \leq n - \sum_{i \in A}(\neg(s_{b+i} \oplus s_{\text{reward},i})) + 1$$

On the other hand, consider the following policy:

$$\pi^*(s) = \begin{cases} 1, & \text{if } s_{b+i} \neq s_{\text{reward},i}, \text{ where } i = \sum_{j=1}^{b} 2^{b-j} \cdot s_j \\ 0, & \text{otherwise.} \end{cases}$$

This policy reaches the reward state in $2^b + \sum_{i \in A}(\neg(s_{b+i} \oplus s_{\text{reward},i})) - 1$ time steps. Indeed, the control bits takes $2^b - 1$ times steps to reach $\mathbf{1}_b$. During these time steps, the control bits (as a binary number) traverses $0, \ldots, 2^b - 1$. Thus for any $i \in [n]$ such that $s_{b+i} \neq s_{\text{reward},i}$, there exists a time $t$ such that the control bits at this time $t$ (as a binary number) equals $i$, and at this time step, the agent flipped the $i$-th coordinate of the representation bits by playing action 1 (note that the flipping operation can always be applied since the condition is always satisfied under the current policy). Therefore, the representation bits takes $\sum_{i \in A}(\neg(s_{b+i} \oplus s_{\text{reward},i}))$ times steps to reach $s_{\text{reward}}$. Combining, the agent arrives the state $(\mathbf{0}_b, s_{\text{reward}})$ after $2^b + \sum_{i \in A}(\neg(s_{b+i} \oplus s_{\text{reward},i})) - 1$ time steps, and then collects reward 1 in each of the remaining $n - \sum_{i \in A}(\neg(s_{b+i} \oplus s_{\text{reward},i})) + 1$ time steps, resulting in a value of

$$V^*(s) = n - \sum_{i \in A}(\neg(s_{b+i} \oplus s_{\text{reward},i})) + 1.$$

$\square$

Now we return to the proof of Theorem 3.8.

*Proof.* First, we show that the model function can be computed by circuits with polynomial size and constant depth. Consider the following circuit:

$$C_{\text{model}}(s,a) = (g(s_1, \ldots, s_b), (s_{b+k} \oplus (\delta_k(s_1, \ldots, s_b) \wedge a \wedge C(s[r])))_{k=1}^{n})$$

where $\oplus$ is the XOR gate, $g : \{0,1\}^{b+1} \to \{0,1\}^b$ such that $g(x_1, \ldots, x_b, a)_i = (f(x_1, \ldots, x_b)_i \wedge \neg a) \vee (x_i \wedge a)$ for all $i \in [b]$, and $\delta_k(x_1, \ldots, x_b) = \mathbb{1}(k = \sum_{j=1}^{b} 2^{b-j} \cdot x_j)$. We can verify that $C_{\text{model}} = \mathbb{T}$. By Claim B.2, Claim B.3, and Lemma B.5, the XOR gate, the control function $f$, the function $C(s[r])$, and the gate $\delta_k$ can all be implemented by binary circuits with polynomial size and constant depth. As a result of Claim B.1, $C_{\text{model}}$ also has polynomial size and constant depth.

Now, the reward function can be computed by the following simple circuit:

$$C_{\text{reward}}(s) = s_1 \wedge \ldots \wedge s_b \wedge (\neg(s_{b+1} \oplus s_{\text{reward},1})) \wedge \cdots \wedge (\neg(s_{b+n} \oplus s_{\text{reward},n})).$$

Finally, we show that with high probability, the value function can not be computed by a circuit with constant depth and polynomial size. By Lemma B.6, with probability at least $1 - e^{-\Omega(m)}$, there exists a set $A \subset [n]$ such that $|A| \geq n/2$ and $C(x) = 1$ holds for any binary string $x \in \{0,1\}^n$ satisfying $x_i = s_{\text{reward},i}, \forall i \notin A$. We can reduce the size of $A$ by deleting some elements to make $|A| = \lceil n/2 \rceil = 2^{b-1} - 1$ and the above property still holds. Denote $A = \{a(1) < \cdots < a(L)\}$ where $a : [L] \to [n]$. Define $h : \{0,1\}^L \to \{0,1\}^b$

$$h((x_1, \ldots, x_L)) = n + 1 - V^*((\mathbf{0}_b, s')), \text{ where } s'_j = \begin{cases} s_{\text{reward},j}, & j \notin A \\ \neg x_{a^{-1}(j)} \oplus s_{\text{reward},j}, & j \in A \end{cases}.$$

Due to Lemma B.7, $h(x)$ can be represented in binary form with $(b-1)$ bits, and the first (most significant) bit of $h(x)$ is the majority function of $x$. If there exists a circuit $C_{\text{value}} = V^*$ with polynomial size and constant depth, then the circuit defined by

$$C_{\text{MAJORITY}}(x_1, \ldots, x_L) = n + 1 - C_{\text{value}}\left(\mathbf{0}_b, \neg x_{a^{-1}(1)} \oplus s_{\text{reward},1}, \ldots, \neg x_{a^{-1}(n)} \oplus s_{\text{reward},n}\right)$$

where $a^{-1}(i) = i$ if $i \notin A$, outputs $h$. By Proposition A.4, $C_{\text{MAJORITY}}$ also has polynomial size and constant depth, which contradicts Proposition A.2. This means that $V^*$ can not be computed by circuits with polynomial size and constant depth. Notice that $V^*(s) = \max_{a \in \mathcal{A}} Q^*(s, a) = \max\{Q^*(s, 0), Q^*(s, 1)\}$. By Proposition A.6 and Claim B.1, we conclude that the optimal $Q$-function can not be computed by circuits with polynomial size and constant depth. $\square$

## C  EXPERIMENT DETAILS

Table 1 shows the parameters used in SAC training to learn the optimal $Q$-function in Section 4. Table 2 shows parameters for fitting neural networks to the value, reward, and transition functions in Section 4.

| Hyperparameter | Value(s) |
|---|---|
| Optimizer | Adam (Kingma & Ba, 2014) |
| Learning Rate | 0.0003 |
| Batch Size | 1000 |
| Number of Epochs | 100000 |
| Init_temperature | 0.1 |
| Episode length | 1000 |
| Discount factor | 0.99 |
| number of hidden layers (all networks) | 256 |
| number of hidden units per layer | 2 |
| target update interval | 1 |

Table 1: Hyperparameters in Soft-Actor-Critic

| Hyperparameter | Value(s) |
|---|---|
| Optimizer | Adam (Kingma & Ba, 2014) |
| Learning Rate | 0.001 |
| Batch Size | 32 |
| Number of Epochs | 100 |

Table 2: Hyperparameters of fitting neural networks to the value, reward, and transition functions

## D  ADDITIONAL EXPERIMENT RESULTS

Under the same experiment settings in Section 4, we conduct further experiments using varying neural network depth and width to approximate the the optimal $Q$-functions, reward functions, and transition functions. The results are shown in Figure 4-Figure 7. Furthermore, since the optimal $Q$-function calculated by the critic in the SAC algorithm might be biased due to under-estimation since clipped double Q learning is adopted, we also use critics of the Actor-Critic algorithm (Appendix D.1) and Monte Carlo simulation to evaluate the $Q$-function of the learned actor (Appendix D.2) to obtain an unbiased estimation of the optimal $Q$-function.

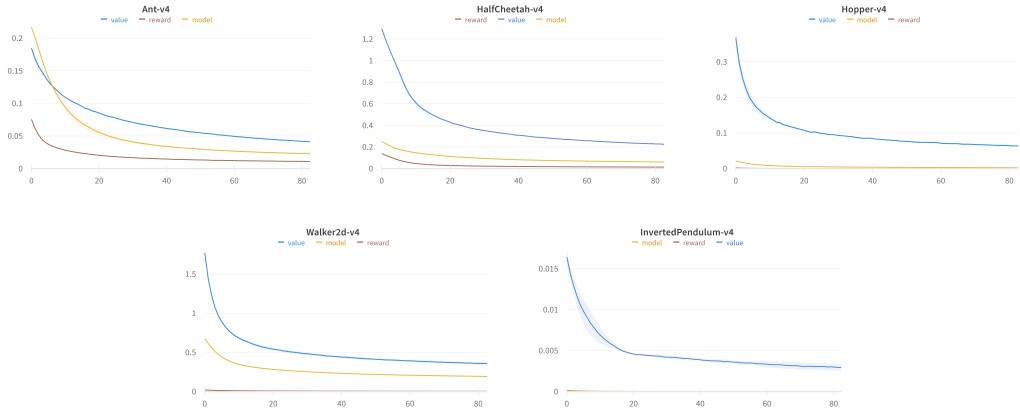

Figure 4: Approximation errors of the optimal $Q$-functions, reward functions, and transition functions in MuJoCo environments, using 2-(hidden) layer neural networks with width 128. In each environment, we run 5 independent experiments and report the mean and standard deviation of the approximation errors.

### D.1  FITTING Q-VALUES LEARNED BY ACTOR-CRITIC

In addition to approximating the Q-functions learned from SAC, we also fit the value functions learned in Actor-Critic algorithm. The result, displayed in Figure 8, exhibits that the value functions are more difficult to approximate than the reward and the transition functions.

### D.2  FITTING Q-VALUES FROM MONTE-CARLO SAMPLING

We also consider Monte-Carlo estimate of the Q-values of the actors learned by SAC. In Figure 9, we visualize the approximation errors (in log scale), which confirms the phenomenon found in Section 4.

## E  EXTENSIONS TO STOCHASTIC MDP AND POMDP

In this section, we briefly discuss possible approaches of generalizing our frameworks to stochastic MDP and partially observable MDP (POMDP).

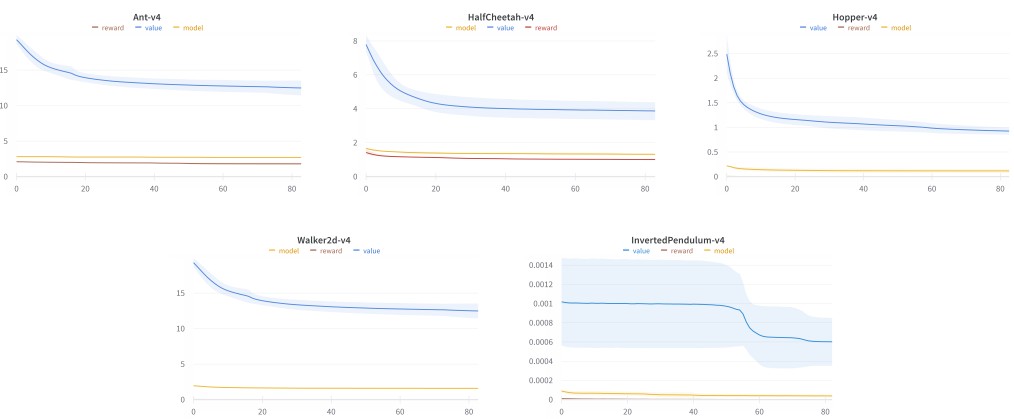

Figure 5: Approximation errors of the optimal $Q$-functions, reward functions, and transition functions in MuJoCo environments, using $1$-(hidden) layer neural networks with width $16$. In each environment, we run $5$ independent experiments and report the mean and standard deviation of the approximation errors.

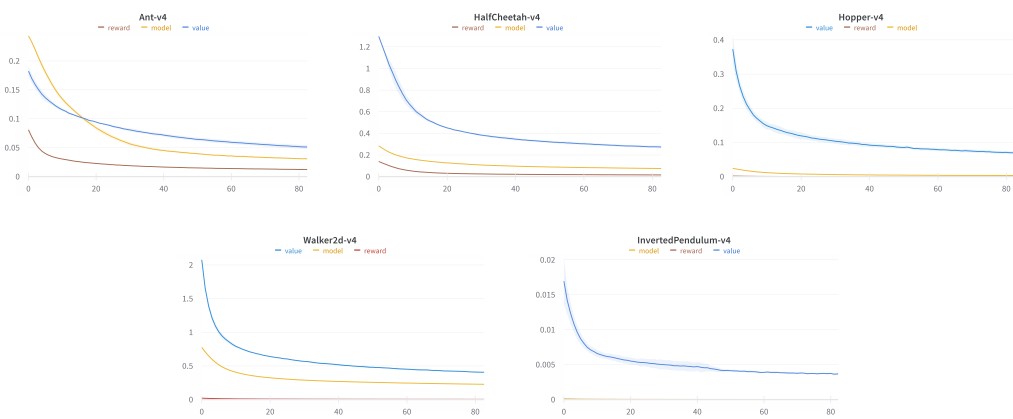

Figure 6: Approximation errors of the optimal $Q$-functions, reward functions, and transition functions in MuJoCo environments, using $3$-(hidden) layer neural networks with width $64$. In each environment, we run $5$ independent experiments and report the mean and standard deviation of the approximation errors.

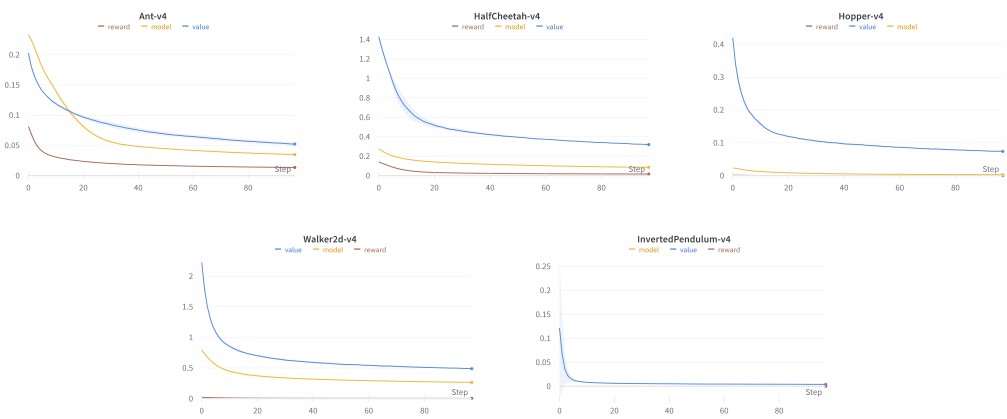

Figure 7: Approximation errors of the optimal $Q$-functions, reward functions, and transition functions in MuJoCo environments, using 2-(hidden) layer neural networks with width $64$. In each environment, we run $5$ independent experiments and report the mean and standard deviation of the approximation errors.

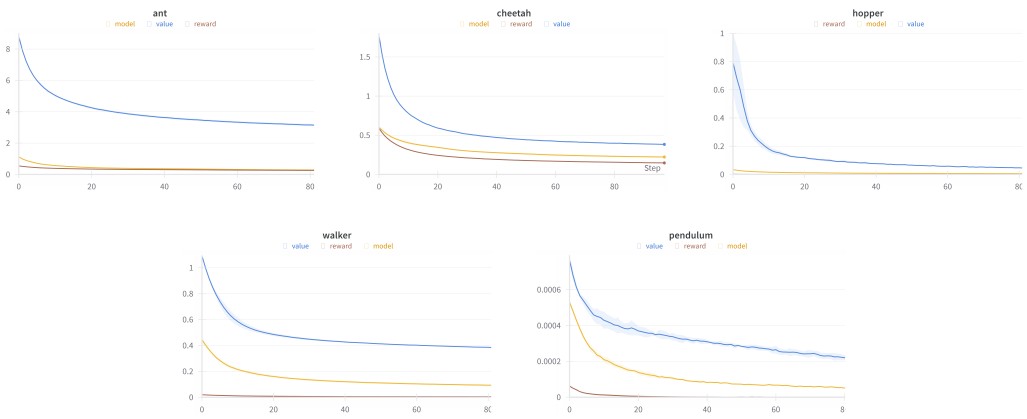

Figure 8: Approximation errors of the optimal value functions (learned by actor-critic), reward functions, and transition functions in MuJoCo environments, using 3-(hidden) layer neural networks with width $64$. In each environment, we run $5$ independent experiments and report the mean and standard deviation of the approximation errors.

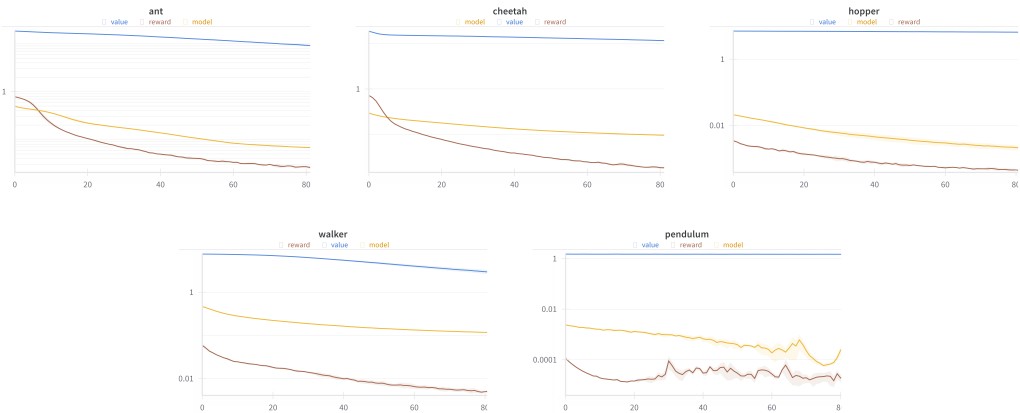

Figure 9: Approximation errors of the optimal $Q$-functions, reward functions, and transition functions in MuJoCo environments, using 2-(hidden) layer neural networks with width 128. For each state, we take 3 Monte-Carlo sampling and use the average discounted sum of rewards as the Q-value estimate. The approximation errors are shown in the log scale. In each environment, we run 5 independent experiments and report the mean and standard deviation of the approximation errors.

**Stochastic MDP.** Although we study the deterministic transition in this paper, it can also be extended to stochastic MDPs. In that case, $P(s, a, s')$ represents the probability of reaching state $s'$ when choosing action $a$ on state $s$. Since the probability is represented with bounded precision in machine (e.g., 64-bit floating point), the output of $P(s, a, s')$ can also be represented by a $\{0, 1\}$-string.

Now we discuss a natural extension from the deterministic transition kernel to the stochastic transition kernel of our majority MDP. After applying $a = 1$, if $C(s[r]) = 1$ (i.e., the condition is satisfied), instead of deterministically flipping the $s[c]$-th bit of representation bits, in the stochastic version, the $s[c]$-th bit will be flipped with probability $1/2$, and nothing happens otherwise. This resembles the real-world scenario where an action might fail with some failure probability. The representation complexity of the transition kernel $P$ remains polynomial. For the optimal $Q$ function, if we change $H = 2^b + 2n$, since in expectation, each (effective) flipping will cost two steps, the representation complexity of the $Q$-function is still lower bounded by the majority function, which is exponentially large. Note that our deterministic majority MDP can be viewed as an embedded chain of a more 'lazy' MDP, and the failure probability can also be arbitrary. Hence, our result (Theorems 3.7 and 3.8) can be generalized to this natural class of stochastic MDPs.

**POMDP.** For POMDP, we can additionally assume a function $O(s) = o$ (or $O(s, a) = o$), which maps a state $s$ (or a state-action pair $(s, a)$) to a corresponding observation $o$. One possible choice of $O(s)$ could be a substring of the state $s$, which still has low representation complexity. In POMDPs, the $Q$-function is often defined on a sequence of past observations or certain belief states that reflect the posterior distribution of the hidden state. As a result, the $Q$-functions in POMDPs often have higher complexity. Since the $Q$-function is proven to possess high circuit complexity for the majority MDP, the representation complexity for value-based quantities of a partially observable version could be even higher.

