# OpenReview forum: "On Representation Complexity of Model-based and Model-free Reinforcement Learning"
_ICLR.cc/2024/Conference — ICLR 2024 poster_

### Official Review · Reviewer_kBCH · 2023-10-27

**Soundness:** 2 fair
**Presentation:** 3 good
**Contribution:** 2 fair
**Rating:** 5
**Confidence:** 2

**Summary:**

This paper studies the representation complexity of model-based and model-free reinforcement learning (RL) algorithms in the context of circuit complexity. The authors introduce a special class of majority MDPs and theoretically demonstrate that while the transition and reward functions of these MDPs have low representation complexity, the optimal Q-function exhibits exponential representation complexity. The paper then extends these findings empirically by examining MuJoCo Gym environments and measuring the relative approximation errors of neural networks in fitting these functions. The results demonstrate that the optimal Q-functions are significantly harder to approximate than the transition and reward functions which supports their theoretical findings.

**Strengths:**

- Novelty: this is the first work to study the representation complexity of RL under a circuit complexity framework. The authors introduced a general class Majority MDPs, such that their transition kernels and reward functions have much lower circuit complexity than their optimal Q-functions. And the authors provides unique insights into why model-based algorithms usually enjoy better sample complexity than model-free algorithms from a novel representation complexity perspective.
- The theoretical results are supported by empirical demonstrations.

**Weaknesses:**

- The theoretical nature of the Majority MDP might limit its direct applicability to practical problems.
- A minor point that the experiment section only includes a limited set of small scale gym environments.

**Questions:**

How might these insights inspire the design and development of improvements to RL algorithms?

---

> ### Author Response · Authors · 2023-11-16
>
> Thanks for reviewing our work and providing helpful and insightful comments. Below are our responses.
>
> >The theoretical nature of the Majority MDP might limit its direct applicability to practical problems.
>
> We discussed in the global response that many MDPs in real-world applications are actually equivalent (or close) to the majority MDP class. Also, our experimental results further validate that our results can be applied to general RL problems.
>
> >A minor point is that the experiment section only includes a limited set of small scale gym environments.
>
> We emphasize that our use of the Mujoco environments aligns with the standard benchmarks prevalent in the reinforcement learning community, as corroborated by multiple studies ([1][2][3]). This alignment not only substantiates our theoretical findings but also ensures their relevance in practical settings. To further fortify the credibility of our experimental results, we provided additional results depicted in Figures 4-7 on pages 20-21 of the revised supplementary material. These enhancements serve to provide a more comprehensive and robust demonstration of our research's applicability.
>
> >How might these insights inspire the design and development of improvements to RL algorithms?
>
> The main message conveyed by our paper that “MDPs with simple model structure and complex Q-function are common’ could inspire researchers to use the knowledge of the model (either using a model-based learning algorithm or boosting existing model-free algorithms using the knowledge of the model or by planning) to achieve better performance than a purely model-free algorithm (as also empirically validated by [4]).
>
> Also, when choosing an appropriate function to learn and deploying function approximation, it is worth thinking about whether the chosen function is simple or complex since to learn a complex function, one either needs a complicated function class or suffers a large approximation error. This is not only important for RL algorithm design but also important for a broader scope of general machine learning settings.
>
>
> **References**
>
> [1] Tuomas Haarnoja, Aurick Zhou, Pieter Abbeel, and Sergey Levine. Soft actor-critic: Off-policy maximum entropy deep reinforcement learning with a stochastic actor. In International conference on machine learning, pp. 1861–1870. PMLR, 2018
>
> [2] Ching-An Cheng, Tengyang Xie, Nan Jiang, and Alekh Agarwal. Adversarially trained actor critic for offline reinforcement learning. In Proceedings of the 39th International Conference on Machine Learning, volume 162, 2022
>
> [3] Hanlin Zhu, Paria Rashidinejad, and Jiantao Jiao. Importance weighted actor-critic for optimal conservative offline reinforcement learning. arXiv preprint arXiv:2301.12714, 2023
>
> [4] Kefan Dong, Yuping Luo, and Tengyu Ma. On the expressivity of neural networks for deep reinforcement learning. In International Conference on Machine Learning (ICML), 2020.

---

> > ### Comment · Reviewer_kBCH · 2023-11-22
> > **thank the authors for the rebuttal**
> >
> > I thank the authors for the rebuttal. I don't have any further questions.

---

> > > ### Author Response · Authors · 2023-11-22
> > >
> > > Thanks for your time reading our response! Would you kindly consider raising your score if we have addressed all your concerns? If not, we are more than willing to discuss more about any unaddressed concerns.

---

### Official Review · Reviewer_zTQ7 · 2023-10-31

**Soundness:** 3 good
**Presentation:** 3 good
**Contribution:** 3 good
**Rating:** 6
**Confidence:** 4

**Summary:**

This work studies the representation complexity of model-based and model-free reinforcement learning (RL). Different from the prior work (Dong et al., 2020b), this paper considers a less restrictive family of MDP and incorporates circuit complexity for more fundamental and rigorous theoretical results. This work introduces the definitions of Parity MDP and Majority MDP and proves that the reward function and the transition function have lower circuit complexity than value functions. Experiments are conducted in MuJoCo continuous control tasks with SAC algorithm, quantitively corroborating the theoretical results.

**Strengths:**

- The paper is well-written and organized. The presentation is smooth and clear.
- To my knowledge, the representation complexity study for the MDP family considered in this paper is novel and significant.
- Both theoretical analysis and empirical investigation are provided.

**Weaknesses:**

- Although the authors mentioned that the considered MDP family is general, it is not clear how the circuit representation complexity in visual-input MDPs (or POMDP) and stochastic-transition MDPs can be. I think at least some discussions and remarks on these points will be useful.
- The experiments are not convincing to me. Some important details are missing, e.g., the exact computation of the approximation error of the optimal Q-function. Moreover, I think the variation in terms of the network scale (i.e., $d$ and $\omega$) and environments (e.g., Atari) is insufficient.

&nbsp;

I would be willing to raise my rating if my concerns are addressed. Please see the concrete questions below.

**Questions:**

1) What are the approximation errors of the optimal $Q$-function calculated exactly in the experiments? Fitting the $Q$ values given by the learned SAC critics or fitting the monte carlo returns of the learned policy (i.e., the actor)?

2) How will the experimental results change when varying the size of neural network used? I think only one configuration (i.e., $d=2, \omega = 32$) is insufficient.

3) How can the circuit representation complexity be in visual-input MDPs (or POMDP) and stochastic-transition MDPs?


==========Post-Rebuttal Comments============

The authors' responses have addressed my questions above. I raised the rating score accordingly.

---

> ### Author Response · Authors · 2023-11-16
>
> Thanks for your effort in reviewing our work and providing helpful and insightful comments. Below are our responses to your questions.
>
> >1. What are the approximation errors of the optimal Q-function calculated exactly in the experiments? Fitting the Q-values given by the learned SAC critics or fitting the monte carlo returns of the learned policy (i.e., the actor)?
>
> In the experiments displayed in Section 4, the approximation errors are calculated by fitting the Q-values given by the learned SAC critics. The Q-values learned by SAC are good approximations to the optimal Q-values. In SAC training, the critic loss (average l2 Bellman error) is consistently bounded by 0.005 in InvertedPendulum-v4, by 0.01 in Ant-v4, HalfCheetah-v4, Hopper-v4, and by 0.1 in Walker-v4 respectively. These small Bellman errors suggest that the Q-functions calculated by the SAC experiments are able to fit the optimal Q-functions quite well. Moreover, we note that the critic losses are all smaller than the (un-normalized)  approximation errors in the Q-function, which are about 10X the relative approximation errors in the Q-function plotted in Figure 3.
>
>
> >2. How will the experimental results change when varying the size of neural network used? I think only one configuration (i.e., d=2,w=32) is insufficient.
>
> Thank you for the valuable suggestion to make the experimental result more solid. As stated in the global response, we conducted further experiments on different configurations of neural networks, and the same result that the approximation errors of the Q-functions are greater than those of the model and reward functions consistently holds under different configurations:
> 1. d=1 w=16
> 2. d=2 w=64
> 3. d=2 w=128
> 4. d=3 w=64
>
> The plots for the experimental results are shown in Figure 4-7, pages 20-21 in the updated supplementary material pdf.
>
> >3. How can the circuit representation complexity be in visual-input MDPs (or POMDP) and stochastic-transition MDPs?
>
> It would definitely be an interesting and important future direction to extend our theories to stochastic MDP and POMDP. Below we briefly discuss possible approaches of generalizing to POMDP and stochastic MDP:
>
> **POMDP**: For POMDP, we can additionally assume a function $O(s) = o$ (or $O(s,a) =o$), which maps a state s (or a state-action pair (s,a))  to a corresponding observation o. One possible choice of $O$ could be a substring of the state s, which still has low representation complexity. In POMDPs, the Q-function is often defined on a sequence of past observations or certain belief states that reflect the posterior distribution of the hidden state. As a result, the Q-functions in POMDPs often have higher complexity.  Since the Q-function is proven to possess high circuit complexity for the majority MDP, the representation complexity for value-based quantities of a partially observable version could be even higher. We thank the reviewer for raising this very interesting question, and a more formal and rigorous analysis would be a fascinating direction for future research.
>
> **Stochastic-transition MDP**: As discussed in the global response, our framework can be extended to stochastic MDP. Here we discuss a natural extension of our majority MDP: After applying a = 1, if C(s[r]) = 1 (i.e., the condition is satisfied), instead of deterministically flipping the s[c]-th bit of representation bits, in the stochastic version, the s[c]-th bit will be flipped with probability 1/2, and nothing happens otherwise.  This resembles the real-world scenario where an action might fail with some failure probability. The representation complexity of the transition kernel $P$ still remains polynomial. For the optimal $Q$ function, if we change $H=2^b + 2n$, since in expectation each flipping will cost two steps, the representation complexity of the $Q$ function is still lower bounded by the majority function, which is exponentially large. Note that our deterministic majority MDP can be viewed as an embedded chain of a more ‘lazy’ MDP, and the failure probability can also be arbitrary. Hence, our conclusion can be generalized to this natural class of stochastic-transition MDPs.
>
> In conclusion, our framework can be extended to more complex settings such as stochastic MDP and POMDP, but the main message of this paper can be effectively and more intuitively conveyed through our majority MDP class. We will add these discussions in the revision. Thank you again for the valuable suggestion.

---

> > ### Comment · Reviewer_zTQ7 · 2023-11-21
> > **Response to Authors' Rebuttal**
> >
> > I appreciate the additional experiments and discussions provided by the authors. Some of my concerns are addressed. And I think the discussions on extending the theories presented in this paper to stochastic MDP and POMDP will be insightful to readers and I recommend the authors to add them in the paper (at least in the appendix).
> >
> > As to the computation of the approximation error of the optimal Q-values, I am still not sure whether using the critics of SAC to be the target to fit is a proper choice. I think MC returns are usually considered to be unbiased in expectation while TD3 and SAC main have underestimation errors (since clipped double Q learning is adopted).

---

> > > ### Author Response · Authors · 2023-11-23
> > >
> > > Thank you for your feedback. In the revised manuscript, we have included further discussions on stochastic MDPs and POMDPs in Appendix E. This elucidates the theoretical underpinnings and practical implications of our work more thoroughly.
> > >
> > > To address your point on the unbiased estimator of the ground-truth Q-function, we have incorporated additional experiment results with Monte-Carlo Q-function estimates in Appendix D.2. We utilized the Monte Carlo (MC) return, averaged over three independent rollouts of the SAC actor, as a proxy for the ground-truth Q-function within the Soft Actor-Critic (SAC) framework. This approach has yielded results that align with our initial findings, where the Q-function's approximation errors are greater than those of the transition and reward functions.
> > >
> > > Moreover, as outlined in Appendix D.1, we leveraged the critic from the vanilla actor-critic algorithms to estimate the ground-truth Q-function. This method is free from the underestimation bias typically associated with clipped double-Q learning, and it also supports our core experimental findings.
> > >
> > > Meanwhile, we are conducting additional experiments with larger Monte Carlo sample sizes and varied configurations to further validate our theorem, which are not reported in this rebuttal due to time constraints. Nonetheless, the preliminary results from experiments D.1 and D.2 corroborate the robustness of our main theorem across diverse real-world scenarios, and we will make sure to incorporate all the experiment results in the future version.
> > >
> > > We hope that these enhancements and clarifications address your concerns. Could we kindly ask you to reconsider your score in light of these revisions? We are fully committed to engaging in further discussions should you have any more questions or feedback.

---

> > > > ### Comment · Reviewer_zTQ7 · 2023-12-05
> > > >
> > > > I appreciate the extra efforts made by the authors during the discussion phase. The additional discussion and resutls addressed my questions. I've raised the rating score accordingly.

---

### Official Review · Reviewer_9wTj · 2023-11-01

**Soundness:** 3 good
**Presentation:** 3 good
**Contribution:** 3 good
**Rating:** 8
**Confidence:** 2

**Summary:**

This paper studies the representation complexity of model-based and model-free reinforcement learning. It proposes to use circuit complexity and proves that for a large class of MDPs, the representation complexity of models is larger than that of Q-values. Furthermore, the paper conducts several experiments showing that the approximation error of the Q-value function class are typically larger than that of the model class, indicating learning Q-value functions is harder than learning models, which coincides with the intuition.

**Strengths:**

1. The paper provides a new perspective of representation complexity in learning MDPs, and the introduction of circuit complexity into this domain seems novel.
2. Both theoretical and experimental results are solid and sound.

**Weaknesses:**

1. While the majority MDPs is broad, it fails to see if common MDPs are in this class or close to this class such that in many real life applications the circuit complexity of the Q-value function class is higher than that of the model class.

**Questions:**

1. While the circuit complexity is a good starting point to study the representation complexity, is it possible that there are other quantities or metrics such that the representation complexity might behave differently?

---

> ### Author Response · Authors · 2023-11-16
>
> Thanks for your support of our work and your time reviewing our paper and providing helpful and insightful comments. Below are our responses.
>
> >While the majority MDPs is broad, it fails to see if common MDPs are in this class or close to this class such that in many real life applications the circuit complexity of the Q-value function class is higher than that of the model class.
>
> We discussed in the global response that many MDPs in real-world applications are actually equivalent (or close) to the majority MDP class. Of course, it is true that we can also construct MDPs of which the model complexity is high while the Q-function is simple, but the main message we aim to convey is that MDPs with simple model structure but complex Q-function are common, and thus using the knowledge of the model (either using a model-based learning algorithm or boosting existing model-free algorithms using the knowledge of the model or by planning) could achieve better performance than a purely model-free algorithm (as also empirically validated by [1]).
>
> >While the circuit complexity is a good starting point to study the representation complexity, is it possible that there are other quantities or metrics such that the representation complexity might behave differently?
>
> We believe that there exist other quantities that can be used to measure the representation complexity, but circuit complexity is one of the most fundamental metrics as the operations in computers are represented by circuits, and this notion is also widely used in TCS.
>
> Note that many complexity measures, such as VC dimension, can be only applied to a function class instead of a single function, and previous attempts, such as using the number of segments of a piecewise linear function [1], are not applicable to more general functions. Instead, the circuit complexity is applicable to a general function (with bounded precision, which is reasonable and actually necessary in practice).
>
> Proposing other reasonable representation complexity measures and studying whether this phenomenon might be different under that measure are definitely interesting future questions for study (We conjecture that the same behavior still happens even under different measures since our experimental results validate this phenomenon using neural networks instead of circuits). We thank the reviewer for raising this point.
>
> **References**
>
> [1] Kefan Dong, Yuping Luo, and Tengyu Ma. On the expressivity of neural networks for deep reinforcement learning. In International Conference on Machine Learning (ICML), 2020.

---

> > ### Comment · Reviewer_9wTj · 2023-11-23
> >
> > Thanks for the responses. I decide to keep my score to encourage the introduction of new concept in the study of complexity in RL.

---

> > > ### Author Response · Authors · 2023-11-23
> > >
> > > We sincerely thank the reviewer again for supporting our work and the great effort in reviewing our paper and providing valuable feedback!

---

### Author Response · Authors · 2023-11-16
**Global response to common questions**

We thank all the reviewers for their helpful and insightful comments. Below we first respond to some common questions.

**Additional experimental results for different neural network architectures**

Thank the reviewers for the suggestion, and we attached additional experimental results for different widths and depths for the neural network structures (see Figure 4-7, pages 20-21 in the updated supplementary material pdf). We additionally choose the following four structures to approximate the transition function P, reward function r, and Q-function:
1. d=1 w=16
2.  d=2 w=64
3. d=2 w=128
4. d=3 w=64

The results for different architectures consistently show that the model dynamics P and r have smaller approximation errors than the optimal Q-function.

**Many real-world MDPs are close to the majority MDP**

The majority MDP is a broad class, and below we argue that many real-world MDPs are actually close to the majority MDP class.

**State Space**: The state space of the majority MDP is the set of all $n$-bit strings. Although the state space for real-world MDPs can be continuous or high-dimensional, to store or represent a state (e,g., in a computer), one will eventually use a {0,1}-string due to the bounded precision. Note that the ‘actual’ state is represented by the representation bits, and the introduction of control bits is to add flexibility to our majority MDP class since we only allow actions to be binary. One can also get rid of the control bits if the action space is enlarged.

**Action Space**:  The action space of the majority MDP is {0,1}. Although the action space for real-world MDPs can be a finite set with cardinality larger than 2 or even continuous, the action will also need to be represented by a m-bit {0,1}-string. Therefore, an m-bit action is equivalent to the composition of m consecutive  {0,1} actions (in that case, we need some additional state bits to store the information of the previous m actions).

**Transition probability P**: Although in this paper we study the deterministic transition, it can also be extended to stochastic MDPs. In that case, P(s,a,s’) represents the probability of reaching state s’ when choosing action $a$ on state $s$. Since the probability is also represented with bounded precision (e.g., 64-bit floating point), the output of P(s,a,s’) can also be represented by a {0,1}-string. We also responded to reviewer zTQ7 regarding their questions for stochastic MDP.
Below we discuss the transition kernel $P$ for our majority MDP.

First, the transition function $f$ for control bits can be an arbitrary permutation, which makes it general since it includes MDPs with all possible orders of operation on representation bits.

Second, our action a=1 only takes effect on one state bit (an atom action), while in many other MDPs, a single action might influence several bits or sometimes almost all of the bits (a complex action). We emphasize that a complex action can actually be accomplished by atom actions. When a complex action might take effect on several (constant number of)  state bits, it can be viewed as a composition of a sequence of consecutive atom actions. When a complex action can influence (almost) all the state bits, e.g., turning off the switch would make the whole room dark, we can introduce an additional bit to represent the status of the switch, instead of actually performing the action on all the state bits.

Also, the transition kernel $P$ depends on the condition function $C(s[r])$, which makes our majority MDP more general and resembles real-world environments. This describes a very general rule that to make some action take effect, certain conditions might need to be satisfied. For example, to kill a dragon, the agent needs to hold a sword or some other weapon; to move towards the east for one step, it must hold that there is no wall in that position.

**Reward function r**: In our majority MDP, there is a reward state where the agent can collect the reward. This is similar to goal-conditioned RL, where the agent needs to reach certain states to get the reward. Note that this is a very broad class of real-world MDPs. Also, for MDPs with multiple reward states, one can add a final state such that each reward state will deterministically transit to the final state and get rewards.

We discuss above that our majority MDP is a broad class, and many real-world MDPs are close to this class. Besides theoretically analyzing the separation of circuit complexity of transition kernel, reward function, and Q-function, our experimental results on real-world MDPs further complement our theoretical results.

---

### Meta-Review · Area_Chair_G97a · 2023-12-06

**Metareview:**

This paper studies the classic question in RL: model-based v.s. model-free learning.

It approaches the question from an interesting new angle of complexity theory, showing that for a family of MDPs that captures many applications well, transition and reward functions can be represent by constant depth circuits with polynomial size while the optimal $Q$-function suffers an exponential circuit complexity in constant-depth circuits.

The use of complexity is quite interesting and inspiring for future works. I recommend acceptance for this paper.

**Justification For Why Not Higher Score:**

NA

**Justification For Why Not Lower Score:**

NA

---

### Decision · Program_Chairs · 2024-01-16

Accept (poster)